# Ionic Conductivity of K-ion Glassy Solid Electrolytes of K_2_S-P_2_S_5_-KOTf System

**DOI:** 10.3390/ijms242316855

**Published:** 2023-11-28

**Authors:** Ram Krishna Hona, Alexa D. Azure, Mandy Guinn, Uttam S. Phuyal, Kianna Stroh, Arjun K. Thapa

**Affiliations:** 1Environmental Science Department, United Tribes Technical College, Bismarck, ND 58504, USA; mguinn@uttc.edu (M.G.); strohkianna@gmail.com (K.S.); 2Engineering Department, United Tribes Technical College, Bismarck, ND 58504, USA; aazure@uttc.edu; 3Environmental Engineering Department, University of North Dakota, Grand Forks, ND 58202, USA; 4School of Arts and Science, University of Mt. Olive, Mount Olive, NC 28365, USA; uphuyal@umo.edu; 5Conn Center for Renewable Energy Research, University of Louisville, Louisville, KY 40292, USA

**Keywords:** glass electrolyte, impedance, ionic conductivity, XRD, ternary composition

## Abstract

Ternary glassy electrolytes containing K_2_S as a glass modifier and P_2_S_5_ as a network former are synthesized by introducing a new type of complex and asymmetric salt, potassium triflate (KOTf), to obtain unprecedented K^+^ ion conductivity at ambient temperature. The glasses are synthesized using a conventional quenching technique at a low temperature. In general, alkali ionic glassy electrolytes of ternary systems, specifically for Li^+^ and Na^+^ ion conductivity, have been studied with the addition of halide salts or oxysalts such as M_2_SO_4_, M_2_SiO_4_, M_3_PO_4_ (M = Li or Na), etc. We introduce a distinct and complex salt, potassium triflate (KOTf) with asymmetric anion, to the conventional glass modifier and former to synthesize K^+^-ion-conducting glassy electrolytes. Two series of glassy electrolytes with a ternary system of (0.9–x)K_2_S-xP_2_S_5_-0.1KOTf (x = 0.15, 0.30, 0.45, 0.60, and 0.75) and z(K_2_S-2P_2_S_5_)-yKOTf (y = 0.05, 0.10, 0.15, 0.20, and 0.25) on a straight line of z(K_2_S-2P_2_S_5_) are studied for their K^+^ ionic conductivities by using electrochemical impedance spectroscopy (EIS). The composition 0.3K_2_S-0.6P_2_S_5_-0.1KOTf is found to have the highest conductivity among the studied glassy electrolytes at ambient temperature with the value of 1.06 × 10^−7^ S cm^−1^, which is the highest of all pure K^+^-ion-conducting glasses reported to date. Since the glass transition temperatures of the glasses are near 100 °C, as demonstrated by DSC, temperature-dependent conductivities are studied within the range of 25 to 100 °C to determine the activation energies. A Raman spectroscopic study shows the variation in the structural units ${PS}_{4}^{3-}{,\;P_{2}S}_{7}^{4-},\;{and\;P_{2}S}_{6}^{4-}$ of the network former for different glassy electrolytes. It seems that there is a role of ${P_{2}S}_{7}^{4-}{\;and\;P_{2}S}_{6}^{4-}$ in K^+^-ion conductivity in the glassy electrolytes because the spectroscopic results are compatible with the composition-dependent, room-temperature conductivity trend.

## 1. Introduction

Potassium batteries (such as the K-ion battery, K-sulfur battery, and K-O_2_ battery) have gained considerable attention in recent years due to their low cost, abundant resources, and the relatively low reduction potential of potassium [1,2,3]. It has been reported that potassium batteries can deliver higher energy-storage capacity and energy density relative to Li- and Na- batteries [2,3]. However, since the liquid electrolyte containing commercially available batteries posed risks of leakage, flammability, and electrode corrosion, attention has been diverted to the development of all-solid-state batteries (ASSBs) [4,5], which are assumed to be the safest energy-storage systems because of the non-flammability and electrochemical stability of solid-state electrolytes [6]. Since battery performance depends on the nature of the electrodes and electrolyte, it is critical to include a high-quality electrolyte and electrodes (cathode and anode) in a K+-ion battery [7]. The introduction of a high-quality solid electrolyte reflects the efficiency of a battery with high energy density, capacity, and durability. For this reason, the search for electrolyte materials with high potassium ion (K^+^) conductivity at room temperature is a crucial aspect in the development of all-solid-state batteries with improved energy and power densities [8].

The increasing research interest in developing all-solid-state K batteries necessitated high room-temperature, K^+^-ion-conducting solid electrolytes. Attempts have been made to develop different types of electrolytes for a solid-state K^+^-ion battery such as solid polymer [9,10] and inorganic electrolytes of different classes and phases [11,12]. Inorganic solid electrolytes have been the focus of investigations [11,12,13]. Most of the studied inorganic solid electrolytes include mixed ionic and electronic conductivity [13,14,15,16]. For all-solid-state potassium batteries (K-ion, K-S, or KO_2_ batteries), a purely K^+^-ion-conducting electrolyte that has no electronic conductivity is required. Among the different classes of purely K^+^-ion-conducting inorganic solid electrolytes, an ionic glass/amorphous electrolyte is one of the potential electrolytes. Ag^+^ and Li^+^ ions conducting glassy electrolytes have proved that the conductivity of a glass electrolyte can be comparable to that required of a liquid electrolyte for room-temperature battery operation [17,18,19]. Glasses are isotropic, without grain boundaries, and are easy to fabricate in the forms of complex shapes that can be used to optimize their properties due to wide compositional flexibility. Much attention has been paid to the study of ionic glasses with Li^+^ and Na^+^ conductivity [20,21,22,23]. However, there has been less attention paid to the study of K^+^-ion-conducting glassy electrolytes. Because of the larger ionic volume of the K^+^ ion relative to Li^+^ and Na^+^ ions, achieving glassy solid electrolytes with high K^+^ ionic conductivity is a major challenge for design and synthesis. Although reports of mixed ions (such as K^+^, Li^+^, Na^+^, etc.) conducting glasses can be found, [24,25] only a handful of reports have studied pure K^+^-ion conducting glass electrolytes [12,26,27,28,29]. Among the five published reports [12,26,27,28,29] we have found, only one report mentioned room-temperature K^+^ ion conductivity that was close to 10^−10^ S cm^−1^ [27].

In general, inorganic alkali ionic glass is composed of a glass network former and glass modifier. Network formers are compounds of a covalent nature such as oxides and sulfides of Si, P, and B, and glass modifiers are compounds of an ionic nature such as oxides and sulfides of alkali metals. Modifiers break long-chain networks to build pathways for cation movement from one spot to another. This makes a glass ion conductive. However, salts are added to improve the ionic conductivity [30] in glasses. Generally, the added salts are alkali metal halides or oxyanion salts of the alkali metals such as MX, M_2_SO_4_, M_4_SiO_4_ (M = Li, Na, and K; x = Cl, Br, and I), etc. [31,32,33,34,35]. Sometimes, nitrides are also added [36]. We are going to introduce potassium triflate, which is a distinct anion and complex in nature. It is assumed that complex anion addition to glass facilitates the paddlewheel mechanism to enhance its ionic conductivity [37]. Here, we report a part of the glass-forming region with different compositions of K_2_S, P_2_S_5_, and KOTf. The conventional alkali ionic glasses were generally synthesized using a melt quench technique at a higher temperature (>700 °C) [20,38]. We report very low-temperature (450 °C) melt-quenched glass systems, (0.9–x)K_2_S-xP_2_S_5_-0.1KOTf (x = 0.15, 0.30, 0.45, 0.60, and 0.75) and z(K_2_S-2P_2_S_5_)-yKOTf (b = 0.05, 0.10, 0.15, 0.20, and 0.25) on a straight line of z(K_2_S-2P_2_S_5_)) along with the ionic conductivity higher than ever reported before for K^+^-ion conducting glassy electrolytes.

## 2. Results

### 2.1. Synthesis and Characterization

TGA analysis showed that KOTf decomposes at around 480 °C (Figure 1). All the compositions of glass electrolytes showed broad or no peaks in the powder XRD, confirming glass phase formation. Two series of glassy electrolytes (each series with five different composition samples) were chosen from the pure glasses for the study of ionic conductivity in the glass system. The studied glass compositions are shown in Figure 2a, and the two series are shown as lines 1 and 2 in Figure 2b. Each series contained a glass composition with an unknown impurity peak. Series 1 contained the compositions of (0.9–x)K_2_S-xP_2_S_5_-0.1KOTf where the amount of KOTf was kept fixed and the ratio of K_2_S and P_2_S_5_ was varied with x = 0.15, 0.30, 0.45, 0.60 and 0.75. The other series (series 2) consisted of the compositions z(K_2_S-2P_2_S_5_)-yKOTf with a ratio of K_2_S and P_2_S_5_ remaining on a straight-line z(K_2_S-2P_2_S_5_) where the z value changes with y.

### 2.2. Ionic Conductivity

In the first series of glasses, (0.9–x)K_2_S-xP_2_S_5_-0.1KOTf, the room temperature conductivity varied with different ratios of glass former to modifier concentrations. The highest conductivity was observed for the composition of x = 0.6, and the lowest conductivity was observed for the composition of x = 0.45. Table 1 shows the room temperature conductivities and activation energies of the 2nd series of glassy electrolytes. In the second series of z(K_2_S-2P_2_S_5_)-yKOTf, the conductivity reached the highest at y = 0.1. Table 2 shows the room-temperature conductivities and activation energies of the 2nd series of glassy electrolytes obtained from the temperature-dependent conductivities measurement from 25 to 100 °C. In the two series, the room temperature conductivities changed with the increase in K^+^ ion concentrations, and the conductivities variations looked opposite. In series one, the conductivity first decreased with the increase in K^+^ ions before the highest conductivity point. However, in the second series, the conductivity first increased and then decreased with the increase in K^+^ concentration, which is just the opposite result to that of the first series.

## 3. Discussion

### 3.1. Synthesis and Characterization

A part of the glass formation region was investigated for a ternary glass system containing K_2_S, P_2_S_5_, and KOTf by the conventional melt quench technique. Before melting the mixture of glass components, we investigated the decomposition temperature of KOTf using TGA and found that it decomposed at around 480 °C (Figure 1). The single-step decomposition of the salt follows the previously reported DSC pattern of other triflate salts of lanthanides and silver [39,40,41]. There was a gradual slope before 480 °C as seen for other reported sulfide materials [42,43]. Since it was accomplished in an open pan and the material was very hygroscopic, the slope before 480 °C was assumed to be due to the loss of adsorbed moisture [42]. We melted and quenched the mixture of precursors at 450 °C, which was below the decomposition temperature of KOTf. In general, the melt quench technique involved higher temperatures, higher than 700 °C [44,45]. Figure 2a shows the pure glass-forming composition with blue circles, and red circles show the formation of impure glass or no glass-forming compositions. Two series of glassy electrolytes (each series with five different composition samples) were chosen from the pure glasses for the study of ionic conductivity in the glass system. The studied glass compositions and the two series are shown in lines 1 and 2 in Figure 2b. Each series contained a glass composition with an unknown impurity peak. Series 1 contained the compositions of (0.9–x)K_2_S-xP_2_S_5_-0.1KOTf where the amount of KOTf was kept fixed and the ratio of K_2_S and P_2_S_5_ was varied with x = 0.15, 0.30, 0.45, 0.60 and 0.75. The other series (series 2) consisted of the compositions z(K_2_S-2P_2_S_5_)-yKOTf with a ratio of K_2_S and P_2_S_5_ remaining on a straight-line z(K_2_S-2P_2_S_5_) where the z value changes with y. The ratio of 1K_2_S-2P_2_S_5_ (or K_2_S/2P_2_S_5_) was chosen for the compositions z(K_2_S-2P_2_S_5_)-yKOTf because the ionic conductivity of this ratio was found to be the best among the studied compositions in the first series. The primary characterization of glass formation or amorphization of a material is generally accomplished by powder X-ray diffraction, as it is easy to distinguish between the crystalline and amorphous phases due to the presence and absence of peaks [46,47]. The glass or amorphous phase shows no peaks or diffused peaks in the X-ray diffraction data [46,47]. We characterized our synthesized materials using powder X-ray diffraction. Figure 3a shows the powder X-ray diffraction patterns of the compositions (0.9–x)K_2_S-xP_2_S_5_-0.1KOTf. All the studied compositions of (0.9–x)K_2_S-xP_2_S_5_-0.1KOTf in series 1 (Figure 3a) produced pure glasses except for the composition with x = 0.45. The vertical lines before 2θ = 30° were observed because of the 3M film. Similarly, all the compositions of z(K_2_S-2P_2_S_5_)-yKOTf in series 2 (Figure 3b) produced pure glass for b = 0.05, 0.10, 0.15, 0.20, 0.25 on a straight line of z(K_2_S-2P_2_S_5_).

The fundamental role of intermolecular interactions on glass formation in multicomponent mixtures can be more complex compared to single-component systems. This complexity arises due to several factors, including the presence of tortuous phase diagrams and the necessity to define the interactions between particles of different types [48]. Past reports of the glass-forming tendency because of the effect of molecular asymmetry have demonstrated that asymmetric molecules can be comparatively more easily supercooled than symmetric ones [48,49]. KOTf is an asymmetric molecule. The study of glassy electrolytes with the addition of KOTf is thus plausible.

The glass transition temperature of a glass can be characterized using digital scanning calorimetry (DSC). Figure 4 and Figure 5 show the DSC data for the series of (0.9–x)K_2_S-xP_2_S_5_-0.1KOTf and z(K_2_S-2P_2_S_5_)-yKOTf. Generally, the glass transition temperature is observed as a slow short slope [50]. However, it can also be observed as a small dip [28,51]. In our case, the glass transition temperature was observed as a small dip, and the temperature corresponding to the first dip has been the glass transition temperature [28]. The glass transition temperature does not seem to change with the compositional variation demonstrating almost no compositional effect on the glass transition temperature. All the materials demonstrated the glass transition temperature near 99–100 °C. A second dip appeared near 140 °C. To confirm the phase transition at the second dip, we heated a pure glass to 150 °C for 2 h and then cooled it down slowly to room temperature to collect its XRD data at room temperature. The XRD data exhibited glass ceramic peaks as shown in Figure 6, confirming the phase transition.

### 3.2. Ionic Conductivity

ASSB requires purely ion-conducting electrolytes. The ionic conductivity of an electrolyte can be determined by EIS studies. We measured the impedance of two series of glasses using EIS within the frequency range of 0.1 Hz to 1 MHz. To determine the highest temperature limit for conductivity measurement at variable temperatures, differential scanning calorimetry (DSC) data were collected for KOTf, which showed the glass transition temperature near 100 °C and another peak near 140 °C. To analyze the peak at 140 °C, XRD data were collected after heating the glass at 150 °C for 2 h, which exhibited peaks (Figure 6) confirming the phase transformation. Glassy electrolytes lose their properties after the phase transition temperature. Since the glass transition temperatures were near 100 °C, the conductivities of the glasses were measured within the temperature range of 25 to 100 °C. Generally, impedance is measured at low voltage (10–100 mV). However, different problems are observed while applying low voltage to measure impedance for a material with very high impedance [52]. So, high voltage is used to measure the high impedance of a solid material [53]. Our materials exhibited the impedance of several GΩ at room temperature. So, we used 20 to 100 mV to measure the impedance of all the studied materials depending on the impedance of the materials. Glass lacks grain boundaries. The ideal EIS of glasses with ionic conductivity exhibits a semicircle. When the impedance of glass is measured as an electrolyte, bulk resistance from bulk material, the capacitive effect of the bulk and the double-layer capacitance at the interface of the electrode and material can be considered. Hence, we used an equivalent circuit for our materials, as shown in the inset of Figure 7 where R1 and R2 represent the resistance of wire (lead) and the bulk material, respectively. CPE1 and CPE2 indicate the capacitance of the bulk material and the electrode–electrolyte double-layer capacitive behavior, respectively [54]. The constant phase element (CPE) is a modeling element frequently used in electrochemical impedance spectroscopy (EIS) to describe nonideal capacitive behavior. It is particularly useful for systems that exhibit imperfect semicircular impedance plots in Nyquist plots. The nonideal behavior is caused by the leakage capacitance, surface roughness, and nonuniform distribution [54]. Semicircles with heights (depressed) smaller than half of their diameter were observed in our materials (Figure 7). Thus, the classical parallel R//C circuit, which consists of a resistor (R) in parallel with a capacitor (C), is not suitable for modeling the impedance behavior of glasses that exhibit imperfect semicircles with the center located on the real axis in the Nyquist diagram. Figure 7 shows an impedance data of 0.3K_2_S-0.6P_2_S_5_-0.1KOTf as a representative of the compositions (0.9–x)K_2_S-xP_2_S_5_-0.1KOTf and z(K_2_S-2P_2_S_5_)-yKOTf. The inset demonstrates the equivalent circuit used and the impedance data at 75 and 100 °C. The circles represent the raw data, and the red solid lines represent the fitted data. The observed curve shows the typical ionic conductivity behavior containing a high-frequency semicircle which represents the bulk properties of the material, and the low-frequency tail is the response of interfacial (electrode) polarization [55].

Some of the studied materials did not show the tail region at room temperature, which might be due to the frequency limitation, as it showed the high resistivity. The semicircle without a spike observation at lower temperatures for some materials is not surprising. It has been mentioned for other materials in the past reports [56,57,58]. The resistance of a material under study is represented by the intercept of the semicircular curve on the low-frequency region. This resistance (R) is used to determine the conductivity (σ) of the bulk material (circular pellet) using its thickness (L) and surface area (A) as given in relation 1 [59].
σ = L/RA(1)

The studies of temperature-dependent conductivity of glassy electrolytes can also characterize their nature. Thus, almost all the glassy electrolytes studied for their ionic conductivities were studied for their conductivities at variable temperatures ranging from ambient to high temperature. The temperature-dependent conductivities of all the studied glasses were measured from 21 to 100 °C. All glasses exhibited the semiconductor-type nature by decreasing resistivity with the increase in temperature within the measured temperature range. The measurement was stopped at 100 °C because the glassy electrolytes showed the glass transition temperature near this temperature. From these experiments, the activation energies were obtained, as shown in Table 1 and Table 2. The conductivity trends and the Arrhenius plots for both series of glasses are shown in Figure 8. The plots in Figure 8b,d were used for fitting with the Arrhenius equation for thermally activated conductivity [59] (shown below), which helped to find the activation energy.
σ T= σ° e ^(−Ea/kT)^(2)
where σ° = pre-exponential factor;

K = Boltzmann constant;T = absolute temperature; andEa = activation energy for conductivity. The activation energy (Ea) can be calculated from the slope of the lines of best fit in the log σT vs. 1000/T plot.

**Figure 8 ijms-24-16855-f008:**
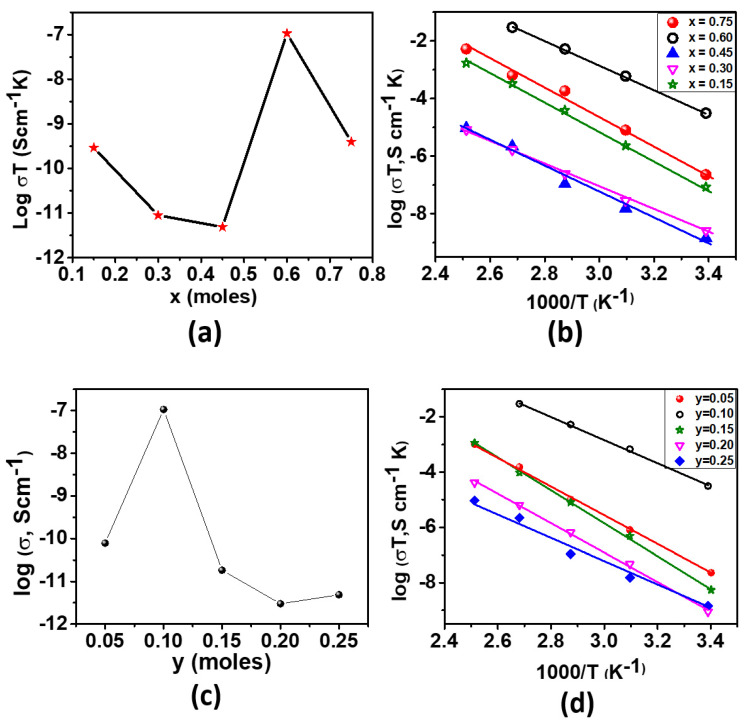
Ionic conductivity of (0.9–x)K_2_S-xP_2_S_5_-0.1KOTf: (**a**) composition dependence and (**b**) temperature dependence and that of z(K_2_S-2P_2_S_5_)-yKOTf (**c**) composition dependence and (**d**) temperature dependence.

### 3.3. Optimizing the K_2_S/P_2_S_5_ Ratio

We studied the effect of the composition of the glassy electrolytes on their conductivities. Most of the ionic conductivity of glassy electrolytes has been studied by varying the ratio of glass former and modifier concentrations [23,60]. We also varied the modifier and network former concentrations in the first series. Figure 8a shows the composition-dependent conductivities of the glassy electrolytes of the first series at ambient temperature. In the first series of glasses, (0.9–x)K_2_S-xP_2_S_5_-0.1KOTf, the room temperature conductivity varied with different ratios of glass former to modifier concentrations. The highest conductivity was observed for the composition of x = 0.6, and the lowest conductivity was observed for the composition of x = 0.45. Since these compositions do not contain any transition elements, the conductivity is believed to be purely due to K^+^ ions. The K^+^ ionic conductivity of x = 0.6 is 1.06 × 10^−7^ S cm^−1^, which is the highest of all the formerly reported values, so far, we have known. The reported values for 0.3K_2_S-0.7(0.1Ga_2_S_3_-0.9GeS_2_) [27] at 25 °C and Ca_3_OK [26] at 75 °C were ~10^−10^ and 10^−13^ S cm^−1^, respectively. Other studies reported K^+^ ion conductivities only above 200 °C [28,29]. Table 1 shows the room temperature conductivities and activation energies of all the studied glassy electrolytes of series 1.

It has been observed in the past in lithium and sodium ion-conducting glasses that the ionic conductivity increases with the increase in alkali ion concentration, which is due to the higher charge carrier concentration [31,61]. This effect has normally been observed in oxide glasses, and the ionic conductivity increases with increasing alkali oxide content. However, there is a concomitant increase in the number of non-bridging oxygens (NBOs) with increasing modifying oxide, which can decrease ionic mobility by acting as trapping centers [23,31]. Some studies reported that the ionic conductivity increased with the increase in glass network former to modifier ratio (GF/GM), indicating the dilution of the charge carrier concentration [23]. Other reports mentioned the highest conductivity at a particular GF/GM ratio [23]. Thus, a fixed rule of the compositional effect of glasses on the ionic conductivity trend has not been established yet. The activation energies of the glasses obtained from the temperature-dependent conductivities also did not show a specific increasing or decreasing order. Thus, the results we obtained in our first series of glasses are not surprising, and the highest room temperature ionic conductivity for (0.9–x)K_2_S-xP_2_S_5_-0.1KOTf is rational.

### 3.4. Optimizing KOTf Concentration

Figure 8c,d show the composition-dependent conductivity and activation energies of the 2nd series of glassy electrolytes with the composition z(K_2_S-2P_2_S_5_)-yKOTf. In this series of glasses, the room temperature conductivity did not increase or decrease systematically with the variation of KOTf concentration. In the Li and Na glassy electrolytes, the Li^+^ and Na^+^ ions of the added salts (MX, M = Li or Na and X = Cl, Br or I) were considered to be adjusted at the interstitial positions [59,62]. Li and Na ion-conducting glasses were found to exhibit increasing conductivity with the concentration of added salts such as MX [38,63]. The increment of the ionic conductivity in those glasses was due to the increase in the charge carrier concentration [38,63]. However, previous reports showed that the conductivity increment stopped at a certain level of added salt concentration, given that the GF/GM ratio remained constant [38,63]. It means that the ionic conductivity becomes the highest at a certain concentration of added salt. In our case of z(K_2_S-2P_2_S_5_)-yKOTf, the conductivity reached the highest at y = 0.1. Table 2 shows the room-temperature conductivities and activation energies of the 2nd series of glassy electrolytes obtained from the temperature-dependent conductivities measurement from 25 to 100 °C. In the two series, the room temperature conductivity changed with the increase in K^+^ ion concentrations, and the conductivity variations looked opposite. In series one, the conductivities first decreased with the increase in K^+^ ions before the highest conductivity point. However, in the second series, the conductivity first increased and then decreased with the increase in K^+^ concentration, which is just the opposite result to that of the first series. If we look back to XRD patterns, the impurity peak appears for a glassy electrolyte after the highest conductivity glass in the first series and before the highest conductivity glassy electrolyte in the second series for K^+^ concentration increment.

### 3.5. Raman Study for Conductivity Trend

Studying the glass structure is most often the key step in explaining or predicting the properties. In general, properties of the alkali oxide and sulfide glasses are described based on the network former (P_2_O_5_/P_2_S_5_) fragment units such as ${PS}_{4}^{3-}{,\;P_{2}S}_{7}^{4-}{,\;P_{2}S}_{6}^{4-}$ [64,65,66]. It is important to study the coordination of the network formers, including their interactions with modifier cations and the added salt. The spectroscopic studies of a previous report for the ${SO}_{4}^{2-}$ addition to Li-based binary oxide glasses have shown that molecular anions can be localized at the interstitial positions of the glass networks [67]. The report also mentioned that the Li^+^ ion sites are affected by the molecular ion, ${SO}_{4}^{2-}$ [67]. However, other reports of SiS_2_ glasses mentioned Li-based sulfide glasses with the formation of new bonds (S-Si-O-Si-S) due to the added complex oxysalts [68]. Similarly, another report mentioned the modification of the glass structure of Li_2_S-SiS_2_ by the addition of Li_3_PO_4_ [69]. Thus, it is hard to conclude whether triflate anion resides in interstitial positions or reacts with the network. Since we are focused here on the K^+^ ionic conductivity, we tried to correlate the role of the dominant coordination structure of the network former with the conductivity of the electrolytes. We discuss Raman spectra of the glassy electrolytes: particularly the highest K^+^ conductive glassy electrolyte, to discuss the structural units. The active bands (peaks) of triflate ions disappear when added to glasses. The reason for the disappearance of Raman peaks of asymmetric triflate anion in glassy electrolytes may be a subject of further investigation, which may help figure out whether the triflate anions reside in interstitial positions or react with the network former to form new bonds. So, we discuss here only the structural units of the P_2_S_5_ fragments which might play a role in the conductivity variation in different compositions of the studied glassy electrolytes. The presence or absence of the fragment units (${PS}_{4}^{3-}{,\;P_{2}S}_{7}^{4-}{,\;P_{2}S}_{6}^{4-}$) were discussed in Li- and Na-ion conducting glasses based on the particular wavenumbers ranging from 380 to 425 cm^−1^ in the previous reports [64,65,66]. According to those reports, the responses of the units ${PS}_{4}^{3-}{,\;P_{2}S}_{7}^{4-}{\;and\;P_{2}S}_{6}^{4-}$ appear as the peaks at 420, 400 and 380 cm^−1^, respectively [64,65,66]. In our glasses also, the peaks appeared at 380, 400 and 420 cm^−1^ corresponding to ${PS}_{4}^{3-}{,\;P_{2}S}_{7}^{4-}{\;and\;P_{2}S}_{6}^{4-}$ units, respectively. Figure 9 shows the Raman spectra of the first and second series of glasses. Figure 9c shows the Raman spectra of the glassy electrolyte (x = 0.6) with the structural unit assignments. As seen for series 1 in Figure 9a, the glassy electrolyte with x = 0.6 composition has a ${P_{2}S}_{6}^{4-}$ unit exactly at 420 cm^−1^. The same peak shifts to the right (higher wavenumber) for x = 0.75. The peak tends to disappear in the composition x = 0.15 where the peak is left as a sign of slope between 400 and 420 cm^−1^. The peak is completely absent in the rest of the two glasses. The order of the disappearance of the structural unit ${P_{2}S}_{6}^{4-}$ is x = 0.60 > 0.75 > 0.15 > 0.30 > 0.45. The room-temperature conductivities also decrease in the same order, σ for x = 0.60 > 0.75 > 0.15 > 0.30 > 0.45. The conductivity trend matches with the appearance or disappearance of the structural unit ${P_{2}S}_{6}^{4-}$. The structural unit ${P_{2}S}_{7}^{4-}$ is clearly observed in the compositions x = 0.6 and 0.75 and partly in 0.15, which is shifted toward the left. This unit is going to disappear in x = 0.30 and is completely absent in x = 0.45, which has the lowest room temperature ionic conductivity. A peak corresponding to ${PS}_{4}^{3-}$ unit can be seen in all the compositions, although it is very small in the composition x = 0.75. A similar trend was observed for the glassy electrolytes of the second series. Figure 9b shows the Raman spectra of the second series of glassy electrolytes where the ${P_{2}S}_{7}^{4-}$ unit trend is 0.1 = 0.05 > 0.15 > 0.20 > 0.25 (peak disappearing from left to right). The conductivity trend is in the order of 0.10 > 0.05 > 0.15 > 0.20 ≈ 0.25. The ${P_{2}S}_{6}^{4-}$ unit is present in all the compositions, but the peak corresponding to y = 0.05 and 0.15 shifted slightly to the right side relative to that corresponding to y = 0.10, and the ${PS}_{4}^{3-}$ peak is absent in the composition, y = 0.25. So, the high conductivity of the glassy electrolytes might correspond to the structural unit ${P_{2}S}_{6}^{4-}$ and ${P_{2}S}_{7}^{4-}$. The correlation of the structural units with the conductivity trends has been reported for other glassy electrolytes in the past [65,66,70]. The correlation of the peak positions (structural units) with the ionic conductivity is opposite to the results reported for Li-ion conducting glassy electrolytes, where a higher conductivity was observed for the glassy electrolyte with a high concentration of ${PS}_{4}^{3-}$ units [65,66]. Some authors reported the high conductivity of glass and glass ceramics as a result of the dominant ${P_{2}S}_{7}^{4-}$ structural units [70,71].

However, the relation has not been found to be uniform for other glasses [72]. It seems that there is no compositional effect on the Tg values of our glassy electrolytes. So, the relationship of room temperature conductivity with Tg values cannot be established for our glassy electrolytes. Some reports mentioned the correlation of activation energy with the conductivity trend. They reported that the high conductivity is due to the low activation energy [33,72,73]. This correlation has been shown to oxide glasses as well [33,74]. In the two series of our electrolytes, the one with the highest conductivity shows the lowest activation energy in the second series and is close to the one with the lowest value in the first series. However, the conductivity is not determined by a single factor; multiple factors are responsible for the variation of the conductivity of glassy electrolytes [33].

## 4. Methods and Materials

Glassy electrolytes were synthesized from three chemicals: K_2_S (99.99%, Alfa Aesar, Haverhill, MA, USA), P_2_S_5_(99.99%, Alfa Aesar) and KOTf (potassium triflate, 99.99%, Alfa Aesar) as starting materials at different compositions. Before their use, all the chemicals were dried at 120 °C for 2 days. The stoichiometric amount of each chemical was weighed for each composition. They were mixed homogeneously in mortar and pestle and kept in a quartz tube, which was sealed later. All the work took place in an argon environment inside a glovebox. The quartz tube was then taken out of the glovebox and vacuum sealed and heated to 450 °C for 6 h. It was quenched in liquid nitrogen. The phase purity and glass formation were characterized by powder X-ray diffraction using a Brucker D8 ADVANCE with an X-ray diffractometer copper Kα-radiation device (1.5418 Å). Since the samples were air sensitive, the materials were covered with 3M film to collect the XRD data. A differential scanning calorimetry (DSC) method was used to characterize the glass transition temperatures. The measurements were carried out in the temperature range from −40 to 200 °C in Al pans with a heating rate of 10 °C min^−1^ in dry argon atmosphere using DSC Q200-1776. The Raman spectra were studied with a 514/613 nm argon laser on a Renishaw inVia Spectrometer. It was scanned from 100 to 800 cm^−1^ with a 50× objective lens. Since the samples are susceptible to moisture, they are sealed in cylindrical glass tubes for spectroscopic data collection. The ionic conductivities of the glass samples were investigated by electrical impedance spectroscopy (EIS) using Gamry reference 600 potentiostat. Impedance measurements were conducted using a computer-controlled frequency response analyzer within the frequency range of 0.1 Hz to 1 MHz. For the measurements, pellets of as-synthesized materials were pressed into a disc of 6 mm diameter using a pressure of 2 tons. The pellets were painted with Pt spray on both sides using plasma sputtering for 4 min on each side. This platinum coating acts as a current collector and electrode. The pellet was kept in an air-sealed Swagelok cell. The whole setting of the pellet in the Swagelok cell was performed inside a glovebox, and the AC impedance measurement was performed outside the glovebox in an MTI 1100 box furnace. The pellet in the Swagelok cell was first sintered at 125 °C for 4 h to bring the particles into close packing, and then the impedance measurement was started from 25 to 100 °C.

## 5. Conclusions

K^+^ ion conducting ternary glassy electrolytes with complex and asymmetric triflate anion as an added salt have been synthesized at low-temperature quenching by a conventional melt quench technique. Ionic conductivities of two series of glassy electrolytes have been studied. The highest RT conductivity was exhibited by a common composition 0.3K_2_S-0.6P_2_S_5_-0.1KOTf of the two series. The highest RT conductivity of 1.06 × 10^−7^ S cm^−1^ is unprecedented for the K^+^ ion conductivity studied yet for glassy electrolytes and is almost three orders of magnitude higher than the highest RT K^+^ ion conductivity reported yet [27]. Based on the structural unit correlation with the composition-dependent conductivity trend, it can be considered that the K^+^ ion conduction is governed by the nature of structural units of the glass former (P_2_S_5_) fragments, particularly by ${PS}_{6}^{4-}$ and ${PS}_{7}^{4-}$.

Overall, our study introduces potassium triflate (KOTf) as a unique and complex salt for synthesizing high-performance glassy electrolytes, thereby paving the way for advancements in all-solid-state potassium batteries.

## Figures and Tables

**Figure 1 ijms-24-16855-f001:**
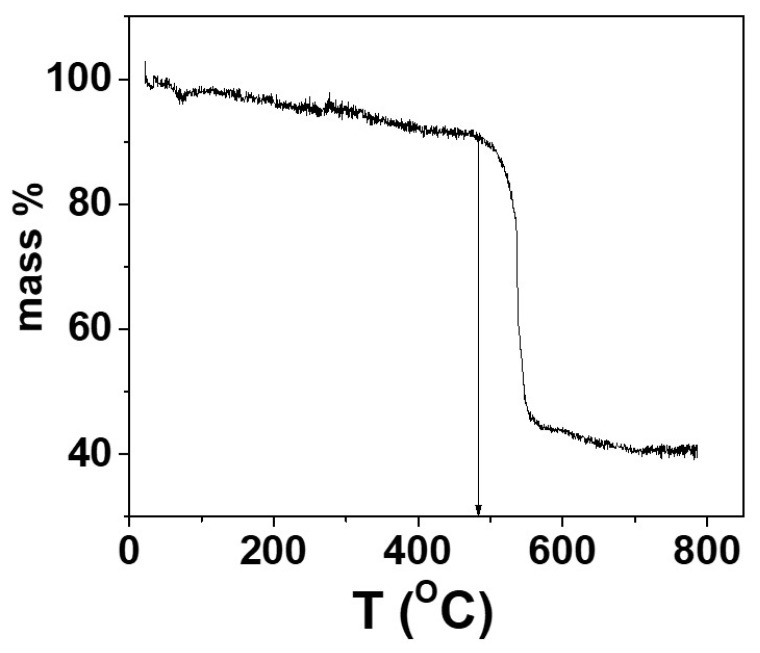
Thermogravimetric analysis (TGA) of KOTf. The decomposition started at around 480 °C.

**Figure 2 ijms-24-16855-f002:**
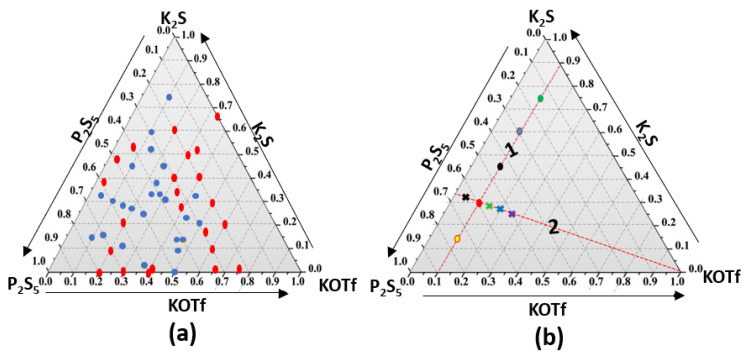
(**a**) Region of glass formation; blue spots represent pure glassy electrolytes and red spots represent impure or no glassy electrolyte formation. (**b**) Glass compositions of (0.9–x)K_2_S-xP_2_S_5_-0.1KOTf (line 1) where x = 0.15 (green), 0.30 (blue), 0.45 (black), 0.60 (red) and 0.75 (yellow) and z(K_2_S-2P_2_S_5_)-yKOTf (line 2) where y = 0.05 (black cross), 0.10 (red circle), 0.15 (green cross), 0.20 (blue cross) and 0.25 (purple cross). The z value changes according to the y values in line 2.

**Figure 3 ijms-24-16855-f003:**
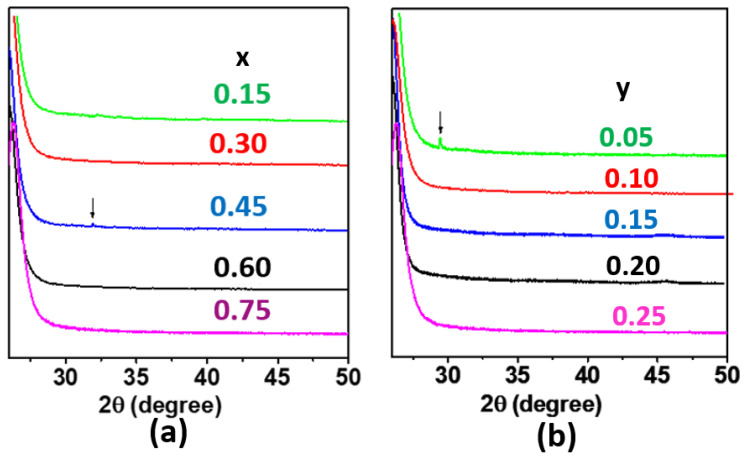
Powder X-ray diffraction data of compositions (**a**) (0.9–x)K_2_S-xP_2_S_5_-0.1KOTf where the numbers represent x values. The arrow on x = 0.45 represents the impurity peak. (**b**) z(K_2_S-2P_2_S_5_)-yKOTf where the numbers represent y values. The arrow on y = 0.05 represents the impurity peak.

**Figure 4 ijms-24-16855-f004:**
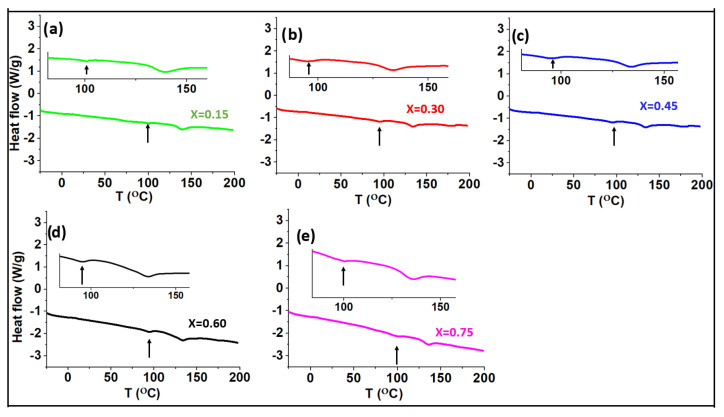
DSC data for the compositions of (0.9–x)K_2_S-xP_2_S_5_-0.1KOTf where x = 0.15, 0.30, 0.45, 0.60, and 0.75. The inset shows the magnified data showing the first dip or slope.

**Figure 5 ijms-24-16855-f005:**
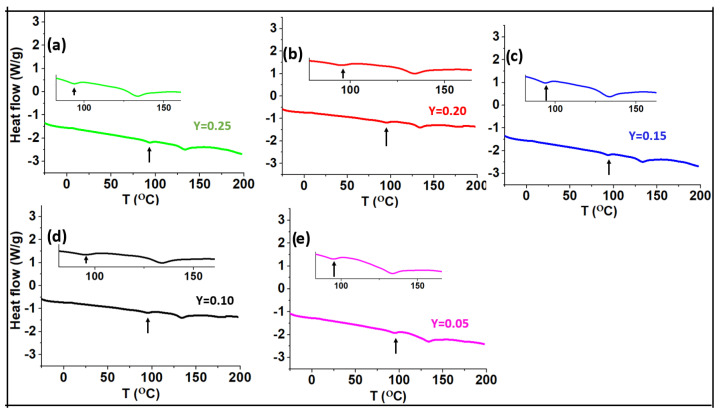
DSC data for the compositions of z(K_2_S-2P_2_S_5_)-yKOTf where y = 0.05, 0.1, 0.15, 0.20 and 0.25. The inset shows the magnified data showing the first dip or slope.

**Figure 6 ijms-24-16855-f006:**
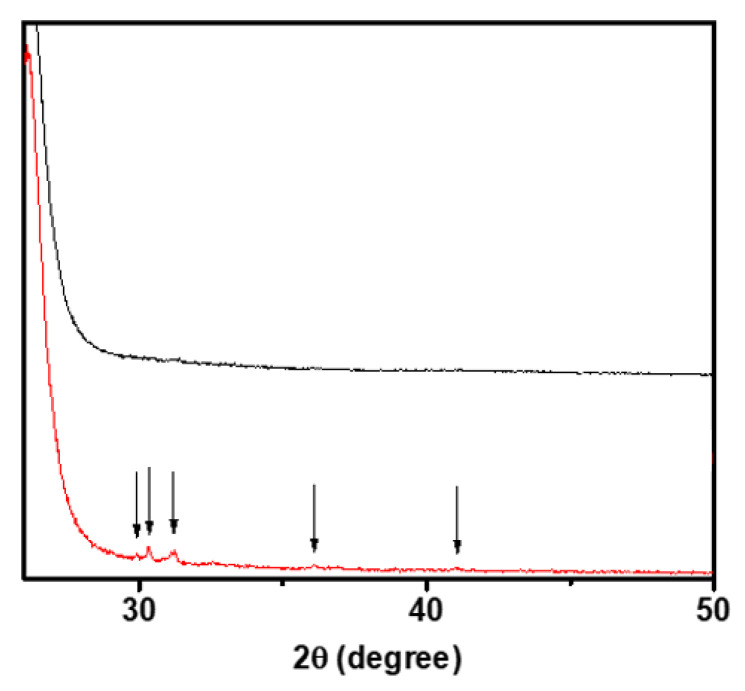
Powder X-ray diffraction data of 0.3K2S-0.6P2S5-0.1KOTf at room temperature. The upper black line represents pure glass. The glass is then heated to 150 °C for 2 h and then cooled slowly. The lower red line containing peaks represents XRD data collected after heating the glass to 150 °C.

**Figure 7 ijms-24-16855-f007:**
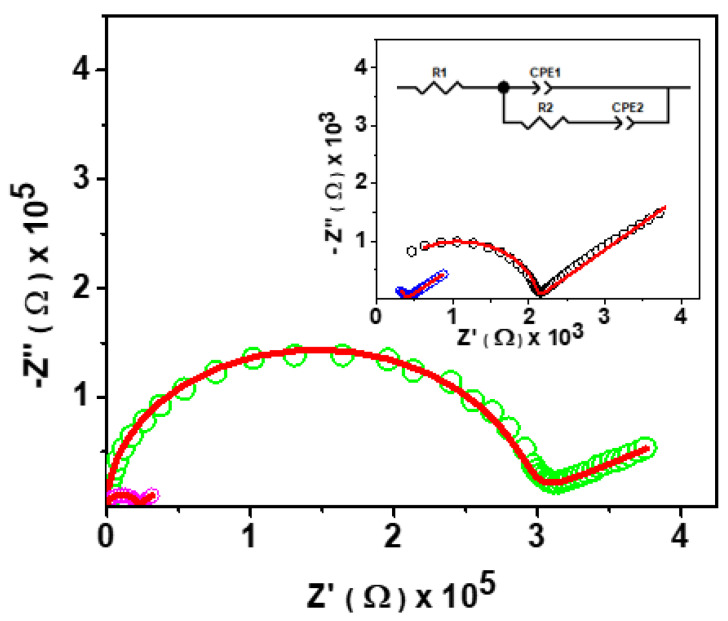
Electrical impedance spectroscopy data of 0.3K_2_S-0.6P_2_S_5_-0.1KOTf. Circles and red solid lines represent raw and fitted data for 25 °C (green), 50 °C (magenta), 75 °C (black), and 100 °C (blue).

**Figure 9 ijms-24-16855-f009:**
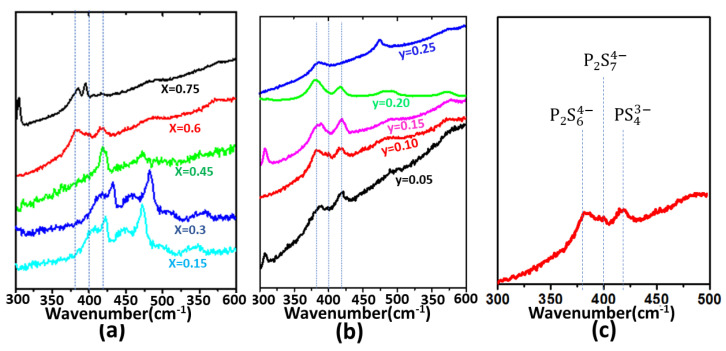
The Raman spectra of (**a**) (0.9–x)K_2_S-xP_2_S_5_-0.1KOTf compositions where x = 0.15, 0.30, 0.45, 0.60 and 0.75; (**b**) z(K2S-2P2S5)-yKOTf where y = 0.05, 0.10, 0.15, 0.20 and 0.25 (**c**) peak assignments for x = 0.60 or y = 0.1.

**Table 1 ijms-24-16855-t001:** Ionic conductivity at 21 °C and activation energy of (0.9–x)K_2_S-xP_2_S_5_-0.1KOTf.

Moles (x)	σ _21_ (S cm^−1^)	Ea (eV)
0.15	2.91 × 10^−10^	0.99
0.30	8.93 × 10^−12^	0.80
0.45	4.88 × 10^−12^	0.89
0.60	1.06 × 10^−7^	0.83
0.75	3.95 × 10^−10^	0.98

**Table 2 ijms-24-16855-t002:** Ionic conductivity at 21 °C and activation energy of z(K_2_S-2P_2_S_5_)-yKOTf.

y	σ _21_ (S cm^−1^)	Ea
0.05	7.81 × 10^−11^	1.04
0.10	1.06 × 10^−7^	0.83
0.15	1.84 × 10^−11^	1.17
0.20	3.00 × 10^−12^	1.05
0.25	3.88 × 10^−12^	0.90

## Data Availability

Data is contained within the article.

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
