# Peer review of "Ionic Conductivity of K-ion Glassy Solid Electrolytes of K2S-P2S5-KOTf System"

_ijms, 2023, doi:10.3390/ijms242316855_

Round 1

Reviewer 1 Report

Comments and Suggestions for Authors

Overall, the manuscript that studies solid state K electrolyte is satisfactory but following need to be improved.

- What is the drying condition, namely normal oven or in a vacuum one?

- At 450C, is the mixture of K2S, P2S5 and KOTf in a liquid form since it is a melt-quench processing?

- What is only thermal gravity analysis of KOTf studied?

- Could not understand well about "different problems are observed to apply low voltage to measure impedance for a material with very high impedance".

- I am interested to learn about EIS of the present electrolyte. It states "The resistance of a material under study is represented by the intercept of the semicircular curve on the low frequency region." So in this case, is there any resistance related to interfaces unless that this piece of electrolyte is pure amorphous? You may wish to check this review although it is for LGPS "Lithium phosphosullide electrolytes for solid-state batteries: Part I, Functional Materials Letters, 15 (2022), 2240001".

- Check if the unit KScm-1 in Fig 8 (a) is correct.

Comments on the Quality of English Language

Can be considered.

Author Response

Please see the response in blue writing in the attached file.

Reviewer 2 Report

Comments and Suggestions for Authors

This is very interesting work, since the problem of selecting an electrolyte is one of the main ones in the design of potassium-ion batteries.

In general, the presented results are interesting and should certainly be published, but I am not sure that this work is suitable for this journal. In addition, the work, in my opinion, needs a little adjustment.

I think the number of analytical methods for interpreting the obtained glasses should be increased. It would be good to conduct SEM and EDX studies to confirm that the glass is homogeneous. FTIR, NMR and synchrotron X-ray diffraction data can also be provided for full interpretation. I understand that not all of these methods may be available to your group, but at least some of them can significantly improve the data.

There are some small comments on the design of the publication, so Fig 1 is too low resolution. Line 144-153 wrong font size, line 237 unnecessary points, Fig 3 (a) and (b) is too big, etc.

Author Response

(The authors gave the same response as above.)

Round 2

Reviewer 2 Report

Comments and Suggestions for Authors

A number of my correction weren't fixed, but overall, after corrections, the text looks complete and worthy of publication.